# Inverse Contrast in Non-Destructive Materials Research by Using Active Thermography

**DOI:** 10.3390/ma12050835

**Published:** 2019-03-12

**Authors:** Paweł Noszczyk, Henryk Nowak

**Affiliations:** Department of Building Physics and Computer Design Methods, Faculty of Civil Engineering, Wroclaw University of Science and Technology, 50-370 Wrocław, Poland; henryk.nowak@pwr.edu.pl

**Keywords:** thermovision, active thermography, thermal contrast, defect detection, location of inclusions, non-destructive testing, materials research, building partition

## Abstract

Background: it is undesirable for defects to occur in building partitions and units. There is a need to develop and improve research techniques for locating such defects, especially non-destructive techniques for active thermography. The aim of the experiment was to explore the possibility of using active thermography for testing large-sized building units (with high heat capacity) in order to locate material inclusions. Methods: as part of the experiment, two building partition models—one made of gypsum board (GB) and another made of oriented strand board (OSB)—were built. Three material inclusions (styrofoam, granite, and steel), considerably differing in their thermal parameters, were placed in each of the partitions. A 7.2 kW infrared radiator was used for thermally exciting (heating) the investigated element for 30 min. The distribution of the temperature field was studied on both sides of the partition for a few hours. Results: using the proposed investigative method, one can detect defects in building partitions under at least 22 mm of thick cladding. At a later cooling down phase, inverse temperature contrasts were found to occur—the defects, which at the beginning of cooling down were visible as warmer areas, at a later phase of cooling down are perceived as cooler areas, and vice versa (on the same front surface). In the transmission mode, the defects are always visible as areas warmer than defect-free areas. Moreover, a quantitative (defect location depth) analysis with an accuracy of up to 10% was carried out using the Echo Defect Shape method. Conclusions: active thermography can be used in construction for non-destructive materials testing. When the recording of thermograms is conducted for an appropriate length of time, inverse contrasts can be observed (on the same front surface).

## 1. Introduction

The material structure of a building unit determines its physical parameters (e.g., its resistance to various external impacts). In the case of existing building structures, one can use either destructive, semi-destructive, or non-destructive tests to assess their structural components. If it is technically possible, it is desirable to carry out non-destructive tests, which do not adversely affect the tested member. A wide range of non-destructive tests is used in construction to investigate the material structure of individual structural components. The applications of most tests are described in the extensive literature on this subject [1,2,3,4]. Thermography, which is also referred to as thermal (infrared) imaging, has been widely used in construction for non-destructive testing purposes for many years [5,6,7,8,9]. It is anticipated that the number of its applications and the quality of thermal imaging surveys will increase in the nearest future [10,11]. A special kind of thermography is active thermography, which consists of thermally stimulation of the tested element and registering its thermal response to the set controlled excitation over time [12]. This method has been successfully applied in materials research to investigate thin elements (up to a few millimeters thick) [13,14,15]. The most commonly used active thermography methods are: lock in thermography (LT), stepped thermography (ST), and phase thermography (PT) [16,17,18]. Stepped thermography consists of thermally exciting (using an external heat source) the tested element with a continuous heat pulse and recording thermograms as the element cools down. There occur two main phases in this method: heating up (or cooling down) and cooling down (or heating up) of the investigated surface. According to the relevant European standard [19] in step of thermography, an energy source (e.g., a halogen or induction lamp) is switched on or/and off at a particular time for excitation. Contrary to pulse thermography, the thermal signature of the defects or of the rear side of the layer already appears during excitation. The image sequence may be analyzed in the time domain or in the frequency domain. Since it is more difficult to detect inclusions in large-sized and massive members, they are less often investigated using this method [20,21,22,23,24,25,26,27,28,29,30,31,32,33,34,35,36,37,38,39,40,41,42,43,44,45,46,47,48]. No experiment in which the time of recording thermograms is considerably extended until a thermal equilibrium between the tested element and the environment is reached. This was reported in the literature on the subject. As part of the present research, elements with a large thermal mass were tested and thermograms were recorded simultaneously in the transmission mode and the reflection mode for three defects made of different materials. A difference in defect detection depending on the distance from which the tested surface was heated up was observed. In most cases, the occurrence of material inclusions in building units is undesirable since it adversely affects the durability of the whole structure. It is vital to locate such material inclusions in order to ensure the safe serviceability of civil structures and their durability. For this reason, an attempt was made to employ active thermography to investigate building partition models and find out if modelled material inclusions in large-sized (a few centimeters thick) members could be located in this way. In the course of the experiment, a new kind of thermal contrast, which has not been described in the literature, was noticed. Building partition models made of oriented strand board (OSB) and gypsum board (GB) were investigated. Material inclusions made of steel, granite, and XPS (extruded polystyrene), i.e., materials markedly differing in their thermal parameters (Table 1), were modelled in each of the two partition models. The tested element was excited by a long-duration heat pulse. The distribution of the temperature field on the surface was registered simultaneously in the reflection mode (the thermal imaging camera situated on the thermal excitation side) and in the transmission mode (the camera situated on the opposite—relative to the thermal excitation—side of the partition). In this way, the material inclusions inside the investigated building partition model were located and identified. Thanks to the recording of the thermograms of the investigated surface over a longer time, an inverse contrast phenomenon (described further in this paper) was observed. In the earlier publications [49,50], the inverse contrast phenomenon was mentioned in connection with the testing of thin (a few mm) elements of a small size (up to 10–20 cm). In addition, in the paper [51], an inverse contrast occurring on different sides of the tested element (the front and rear side of the material sample) was mentioned. In the present research, the inverse contrast phenomenon occurs on one (the same) side of the tested building partition after a long observation time. The principal aim of this research was to examine the effect of the type of partition materials and the type of material inclusions for the possibility of detecting the latter in massive building partitions, i.e., elements with a large thermal capacity.

## 2. Materials and Methods

Two building partition models, consisting of four 1250 × 1250 mm (oriented strand and gypsum) boards bolted together, were investigated. The surface (outer) boards were 22 mm thick, while the inner boards were 10 mm thick. The overall thickness of the partition model amounted to 64 mm. Materials markedly differing in their thermal parameters, i.e., steel, granite, and styrofoam (XPS), were placed inside the model (inserted into the 10 mm thick boards). The inclusions were 200 × 100 × 20 mm in size and spaced from one another at a distance greater than their largest dimension in order to eliminate the influence of one inclusion on another during heat conduction in the course of the experiment. This distance was estimated on the basis of previously carried out numerical studies of heat conduction in the partition model. A precise geometrical model of the tested element, which shows the arrangement of the material inclusions inside the partition, is presented in Figure 1.

Two thermal imaging cameras are placed on the two sides of the partition model. A thermal excitation source and an air temperature and humidity sensor were used in the experiment. A Flir P65 camera with an infrared (IR) detector resolution of 320 × 240 pixels (the camera on the unheated side of the partition—the transmission mode) and an Optris PI400 camera with an IR detector resolution of 382 × 288 pixels (the camera on the side heated by the infrared radiator—the reflection mode) were used in the experiment. The thermal sensitivity of the two infrared cameras was below 80 mK. A Fobo infrared radiator with a total power of 7.2 kW (six lamps, each with a power of 1.2 kW) was used for thermal excitation. In order to prevent the tested element’s edges from cooling (which would disturb the temperature field there), a Styrofoam band was placed around the partition model. A schematic of the test stand is shown in Figure 2.

As part of the experiment, two building partition models, one made of OSB and the other of GB, were investigated. The experiment consisted in thermally exciting the partition with a 30-minute long continuous heat pulse. The heating was effected at two different infrared radiator distances from the excited surface, i.e., 1500 and 500 mm. The heating of the investigated surface was followed by its cooling during which the temperature field distribution was cyclically measured every 20 s on both sides of the tested element (the transmission mode and the reflection mode). The thermal imaging measurement was conducted for up to eight hours, counting from switching the radiator off. A simple analysis based on the obtained thermograms was carried out. For this purpose, the absolute contrast was calculated from the formula below.
C_a_(t) = T_p_(t) − T_pj_(t)    [°C](1)
where: Tp(t)—the temperature on a surface point of the cross-section with a defect [°C],Tpj(t)—the temperature on a surface point of the homogeneous cross section without a defect [°C].

Measuring points in the thermograms were situated in the center of the area with a defect and at the center of the tested element in the defect-free cross-section. The thermogram analysis software “in-point reading” function was used to read temperature values. The size of the in-point function area was 3 × 3 pixels.

Various building materials were used in the experiment. Besides the kind of thermal excitation and the tested element geometry, the thermal parameters of the materials used also influenced the distribution of the temperature field. The thermal parameters of the materials used in the experiment are specified (on the basis of various literature sources) in Table 1.

The thermal parameters of the tested materials affected the test results the most. The homogeneous materials and the material inclusions were appropriately matched in order to compare the influence of the differences between the thermal parameters on the temperature field distribution recorded during the experiment.

## 3. Results

### 3.1. Thermograms

The obtained temperature field distribution results are presented as thermograms recorded at characteristic instants. Figure 3 shows the results for the measurement performed in the reflection mode with the heating from a distance of 0.5 m for the homogeneous OSB material (row I) and the GB material (row II), and in the transmission mode at the infrared radiator-surface distance of 1.5 m for the GB partition model (row III) and the OSB partition model (row IV) as well as for the measurement performed in the reflection mode with heating from a distance of 1.5 m for the GB partition (row V) and the OSB partition (row VI). The thermogram recording time, counting from the beginning of the element cooling down, is specified above each of the columns (A–D). The numbers from 1 to 18 indicate the temperature reading positions at the center of the cross section with a defect, while “R” designates the temperature reading position at the defect-free cross section.

### 3.2. Absolute Contrasts

On the basis of the obtained temperature field distribution on the investigated surface over time, the absolute contrasts were calculated from Formula (1). The absolute contrast values were calculated in the thermogram points designated with numbers from 1 to 18—the cross sections with defects—whereas “R” designates the temperature value in the defect-free cross section (see Figure 3). Figure 4, Figure 5, Figure 6, Figure 7, Figure 8 and Figure 9 correspond to the next rows, according to Figure 3 (rows I–VI), for the tested partition model surfaces. The characteristic point numbers 1–18 and the cross section designations A–D are the same in Figure 3 and Figure 4. The thick trend line was plotted as a 6th degree polynomial.

In order to illustrate better and compare the differences in absolute contrast values for the particular types of material inclusions, the obtained data are presented in individual diagrams for styrofoam (Figure 10), granite (Figure 11), and steel (Figure 12).

## 4. Discussion

### 4.1. Absolute Contrasts and Absolute Inverse Contrasts

The inclusions incorporated into the OSB and GB partition models, which is invisible from the outside, can be located using the active thermography method. The different thermal properties of the inclusions in comparison with those of the basic material in which these inclusions occur cause disturbances in the temperature field distribution visible in the thermograms. The temperature is uniform in those places where there are no inclusions. The only non-uniformities, which can occur there, are due to the non-uniform heating of the surface (when the surface is heated for up to 60 min from a small distance—Figure 3, rows I and II). In the cross sections with inclusions, one can observe warmer or cooler areas in comparison with the homogeneous cross sections. In the case of measurement in the reflection mode, for both the partition materials (OSB and GB), the Styrofoam inclusion was visible as an area with a higher temperature, whereas the granite inclusion and the steel inclusion were observed as areas with a temperature lower than that of the areas without an inclusion. This dependence was observed during heating from both the distance of 0.5 m and 1.5 m. In the case of heating from the distance of 0.5 m for 30 min, the investigated surfaces was heated up to a temperature of about 90 °C, whereas, for heating from the distance of 1.5 m, the temperature on the surface did not exceed 66 °C. In both cases, the inclusions were clearly visible and locatable, but at surface temperatures, when higher than 90 °C, the investigative method is no longer non-destructive since the excessively high temperature damages the structure of the tested materials. Moreover, in order to reach such a high temperature of the surface, the heat source must be placed close to this surface, which results in the non-uniform heating of the latter, consequently, and reduces the visibility of the inner inclusions. When the surface is excited more intensively, the obtained maximum absolute thermal contrasts range from about 3 °C to over 6 °C (Figure 4 and Figure 5). At less intensive heating of the surface, the maximum temperature contrasts range from about 1 °C to over 4 °C (Figure 8 and Figure 9). As one can see in Figure 3, regardless of the degree to which the surface is heated up, at the 30th min from the beginning of the cooling down phase, all the material inclusions are clearly visible and locatable. Hence, one can conclude that, for this geometrical configuration of the building partition and the material inclusions, it suffices to heat up the surface to a temperature of about 60 °C (using the 30-min long continuous heating by an infrared radiator) in order to locate defects situated under a 22-mm thick layer of the basic material. With regard to the thermal parameters of the materials (Table 1), the two basic materials (OSB and GB) have similar heat capacity, but GB is characterized by a higher thermal diffusivity at nearly twice the value. This means that heat will propagate faster in the partition model made of GB. Styrofoam has a very low heat capacity in comparison with that of the partition material, whereby this cross section heats up very quickly and is visible as an area with a temperature higher than that of the defect-free area. By contrast, granite and steel have a heat capacity much higher than that of GB and OSB, which means that the cross section containing such inclusions heats up more slowly than the homogeneous area around the inclusions. The temperature distribution described above is visible in the reflection mode for both strong and weak heating, up to approximately the 90th min from the beginning of cooling down. After this time, inverse contrasts were observed, i.e., the areas with inclusions, which, at the beginning are visible as cooler areas, and, in the second phase of cooling down (after approximately the 90th min), begin to be visible as warmer areas and vice versa. This phenomenon is visible due to the very long recording of the cooling down of the tested elements. It occurs as a result of the difference in heat capacity between the inclusion and the basic material: steel and granite characterized by the high heat capacity cool down much slower than the partitions made of GB and OSB, respectively. In the case of Styrofoam, due to its lower heat capacity, the cross section with a defect cools down quicker than the homogeneous cross section without a defect. In the experiment, the inverse contrast phenomenon can be best observed for the steel inclusion and the granite inclusion when the GB partition model is heated from a distance of 0.5 m (Figure 3 row II). The change of the absolute contrasts over time is shown in Figure 4, Figure 5, Figure 6, Figure 7, Figure 8 and Figure 9. The sharpest contrasts (the most clearly visible areas with an inclusion) would occur from approximately the 15th to approximately the 40th min since switching the heat source off. For the excitation from a distance of 0.5 m (for both the OSB partition (Figure 4) and the GB partition (Figure 5)) the contrast reached about +4 °C while the cross sections with the steel defect and the granite defect induced a contrast amounting to about −6 °C and about −4 °C, respectively (the minus sign indicates that the defect is visible as a cooler area, whereas the plus sign represents an area warmer than the surrounding area without a defect). In the case of heating from a distance of 1.5 m, the absolute contrast was lower by about 3 °C for the Styrofoam and by about 2 °C for the steel and the granite. Contrasts were also observed during the recording of thermograms in the transmission mode (Figure 3, rows III and IV). The values of the contrasts were considerably lower and amount to ±1 °C (Figure 6 and Figure 7). Even at such low contrasts, however, the defects could be located up to the 60th min from the beginning of the cooling of the partition model.

Absolute contrasts occurring as the element cools down are marked in the diagrams (Figure 13, Figure 14 and Figure 15). In the initial cooling down phase, a normal absolute contrast (positive for material inclusions with low heat capacity and negative for material inclusions with high heat capacity in comparison with the basic partition material) occurs. In the next cooling down phase, inversion of the contrasts occurs. This is referred to as an inverse absolute contrast. The inverse absolute contrasts are much smaller than the normal absolute contrasts, but they still allow the location of defects within the tested partition, even a few hours after the thermal stimulation of the surface has ended. In the adopted geometric partition model, an interesting dependence was observed. In the case of materials with high heat capacity (granite and steel), the contrasts have the same sign in both the reflection mode and the transmission mode, whereas, for defects with low heat capacity (Styrofoam), the contrast is positive in the reflection mode and negative in the transmission mode. Regardless of heat capacity, in the transmission mode, the defects were always perceived as areas warmer than the homogeneous areas without defects.

### 4.2. Defect Location Depth

In addition, other parameters are used in defect detection. In References [16,17], it was noted that parameters “m” and R^2^ could be used for this purpose.

In the above publications, parameter “m” is defined as the inclination and R^2^ as a cooling curve determination coefficient. The cooling curve plot is a Log-Log graph. Time axis X represents Log(t) while axis Y represents the contrast logarithm Log(T − T_0_), where T is the temperature varying over time and T_0_ is the temperature measured before the stimulation. For these assumptions, Figure 16 and Figure 17 showed the determined values of parameters “m” and R^2^ for the Log-Log graph. In Reference [16], when testing a thin plate (16.2 mm) containing defects with a diameter of a few mm, located at a depth of a few mm (the first cycle of testing using the lock-in method was analysed), the determined R^2^ parameter values were in the range of 0.92 to 0.98 (for the selected defects) and the deflections ranged from −0.43 to −0.76. One can notice that both when small elements with a low thermal capacity and massive partitions with a high thermal capacity are tested, similar temperature-time dependences are obtained in the characteristic cross sections with and without defects. A better fit of the cooling line was obtained for the partition model made of gypsum board (R^2^ ranging from 0.96 to 0.99). A better cooling line fit means a greater possibility of locating defects on the thermograms. This dependence is visible in Figure 3 where defects in the GB partition are more distinct and their edges are “sharper” in comparison with the partition model made of OSB.

The echo defect shape (EDS) method can be used to determine the depth at which a defect is located in the tested material. This method is described in References [49,50] in more detail where the depth of defect location is calculated using Equation (2).
(2)$$d=\sqrt{-\frac{\lambda \cdot t}{\rho \cdot c_{w}}\cdot ln\left(C_{rel}\left(t\right)\right)}\;\;\;[m]$$
where:
λ—the thermal conductivity of the tested material (above the defect) [W·m^−1^·K^−1^],t—the time elapsed since the end of heating [s],ρ—the bulk density of the tested material (above the defect) [kg·m^−3^],$c_{w}$—the specific heat of the tested material (above the defect) [J·kg^−1^·K^−1^],$C_{rel}\left(t\right)$—the relative contrast in relation to the reference area (the cross section with the defect) [-].

The relative contrast is a ratio of the absolute contrast to the temperature in a surface point of the homogenous cross section without a defect. When calculating “d”, it is highly important to select a proper relative contrast, which must be high enough to reduce the measurement noise, but still be below the maximum values [50]. In Reference [49], it is recommended to adopt $C_{rel}\left(t\right)$ equal to 0.025 if the thermal noise is in the range of 0.01–0.015. In Reference [50], this value is optimized and selecting the value of 0.07 is recommended. It sometimes happens that the relative contrast does not reach the value of 0.07 during the test. For testing elements with a large thermal mass by means of a long thermal pulse, the authors optimized $C_{rel}\left(t\right)$ value selection to the value that occurs 3 min before the first peak of this contrast. The relative contrast and defect location depth versus time dependence (in the GB partition model heated from a distance of 0.5 m) is shown in Figure 18, Figure 19 and Figure 20.

For the above assumptions, the defect location depth was calculated for two partition models thermally stimulated from different distances. The GB and OSB values contained in Table 1 were used in the calculations. The results are shown in Table 2. The actual depth at which the defects are located amounts to 22 mm. For the presented investigative methodology, the calculated values do not exceed the relative error of 10% (except for the Styrofoam defect in the OSB model, heated from the distance of 1.5 m. In this case, the relative error amounts to 23%).

### 4.3. Dimensions of the Defect

The measurement by means of the Flir P65 camera with a 24° × 18° lens was carried out from a distance of 4.0 m, which means that the size of a single pixel in the thermogram is 5.2 mm. For the Optris PI400 thermal imaging camera with a 60° × 45° lens and measurement taken from a distance of 2.0 m, the pixel size is 4.6 mm. Using the thermograms, one can easily assess qualitatively the location of the material inclusions. A disturbance in the temperature field distribution indicates the place where the inclusions occur. On the basis of the thermograms, one can also carry out a simple quantitative assessment and estimate the dimensions of the defects inside the tested partition. The procedure for estimating the width and height of a defect is schematically shown in Figure 16. First, one should select the thermogram in which the defect is most clearly visible. The best thermogram is the one in which the thermal contrast between the area with the defect and the defect-free area as well as the potential edges of the defect is the sharpest (most distinct). Then, virtual lines representing the predicted edge of the defect are assumed. In the program for analyzing thermograms, one can read the x, y coordinates of the particular lines and, on this basis, calculate the potential dimensions (in pixels) of the perceived defect. Lastly, knowing the size of a single pixel, one can estimate the size of the material inclusion. Figure 21 shows how the defect made of granite is estimated using thermograms obtained from the Optris thermal imaging camera. The estimated dimensions are 211.6 mm × 124.2 mm, while the actual dimensions of the defect are 200 mm × 100 mm. It can be assumed that the relative estimation error does not exceed 6% for the width and 25% for the height. If the above procedure is used, the obtained dimensions will always exceed the actual dimensions of the defect since the assumed area delimiting the place where defects occur is an area of the temperature field disturbance. This disturbance always decays from the defect’s edge outwards instead of inwards. Quantitative procedures for estimating defect dimensions by using pulsed thermography can be found, for example, in References [52,53].

## 5. Conclusions

From the experiment carried out on the building partition with modeled inclusions of various materials as well as from the analysis of the obtained thermograms and the calculated absolute contrasts, the following conclusions can be drawn (for the experimental geometry of the partition and the inclusions).
Using the non-destructive active thermography technique, one can locate material inclusions in building partitions (elements with a large thermal capacity), situated at a depth of at least 22 mm below the surface of cladding made of wood or gypsum;In order to locate defects in a building partition, it is sufficient to heat its surface to about 60 °C whereby the test does not lose its non-destructive character;The highest temperature contrasts between the cross sections with defects and the cross sections without defects (the inclusions are most clearly visible then) arise at the time interval from the 15th to the 40th min since the beginning of cooling down;During testing in the reflection mode, inclusions with a considerably higher heat capacity than that of the basic materials are visible as areas whose temperature is lower than that of the areas without inclusions;During testing in the reflection mode, inclusions with a considerably lower heat capacity than that of the basic materials are visible as areas whose temperature is higher than that of the areas without inclusions;During testing in the transmission mode, the areas with defects are visible as areas whose temperature is higher than that of the areas without inclusions, regardless of the heat capacity of the material inclusions;Approximately 90 min after the heating source is switched off, the second phase of cooling down takes place. In this phase, an inverse absolute contrast appears—the sign of the contrast changes and the inclusions initially visible as a warmer area are perceived as cooler areas, and vice versa (this phenomenon occurs on the same side of the partition);Using the Echo Defect Shape method, one can successfully (with an error below 10%) determine the depth at which a defect is located in the building partition;Knowing the specifications of the thermal imaging camera lens (the size [in mm] of 1 pixel in the thermogram), one can estimate the size of the defects present inside the partition by indicating the place in the thermogram where the defect’s edges are likely situated.

The experiment shows that active thermography has great potential for materials testing in construction. It has been demonstrated that defect detection can be successfully conducted over large areas of massive members (large thermal capacity), using an appropriately powerful heat source and a long heat pulse. By recording the distribution of the temperature field during the cooling down of the tested partition over a long time, inverse contrasts (on the same front side of the partition) were observed. As part of further research, inverse heat conduction problems should be more broadly solved and the depth at which inclusions occur, as well as their thermal parameters, should be estimated on the basis of the obtained test results (thermograms). The further research will deal with applying inverse contrasts to improve the defect detection method (quantitative analysis). It is anticipated that the proposed investigative technique will be increasingly used for the non-destructive testing of materials, especially large-sized building units.

## Figures and Tables

**Figure 1 materials-12-00835-f001:**
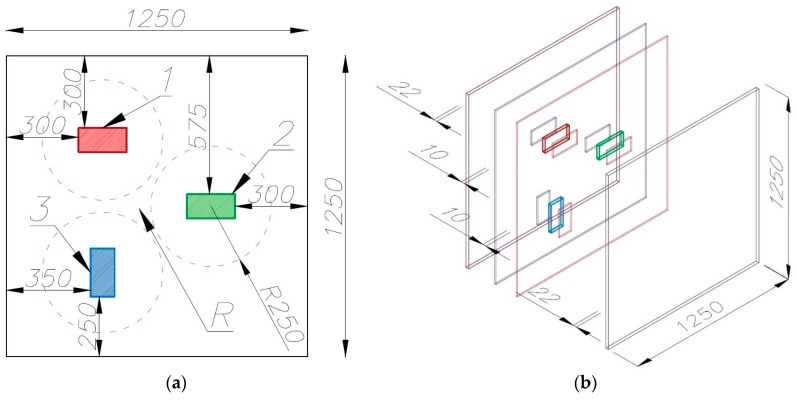
Geometrical structure of the partition model (given units in mm): (**a**) arrangement of material inclusions, internal structure. (**b**) View of the 3D partition model where inner boards with holes and inclusions are marked blue and red, while the outer boards are marked violet and grey. Designations in figures: 1—XPS inclusion, 2—granite inclusion, 3—steel inclusion, and R—homogenous area without inclusion.

**Figure 2 materials-12-00835-f002:**
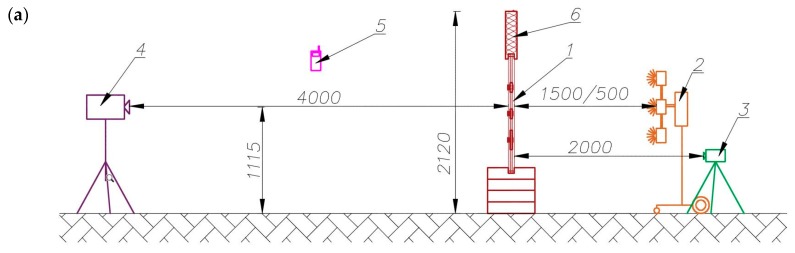
Schematic of test stand (given units in mm): (**a**) side view and (**b**) top view. Designations in the figure: 1—tested partition model, 2—infrared radiator, 3—thermal imaging camera Optris PI400, 4—thermal imaging camera Flir P65, 5—air temperature and relative humidity sensor, 6—Styrofoam band.

**Figure 3 materials-12-00835-f003:**
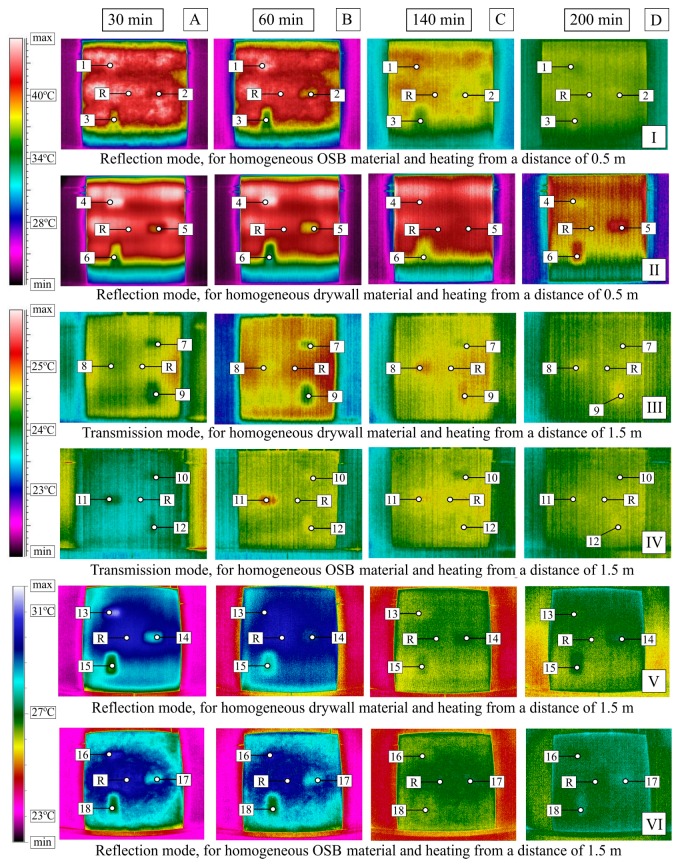
Thermograms at selected instants (time from the beginning of the tested element cooling down is specified above the figure). Numbers from 1 to 18 indicate the temperature reading position at the center of the cross section with a defect, while “R” designates the temperature reading position at the defect-free cross section.

**Figure 4 materials-12-00835-f004:**
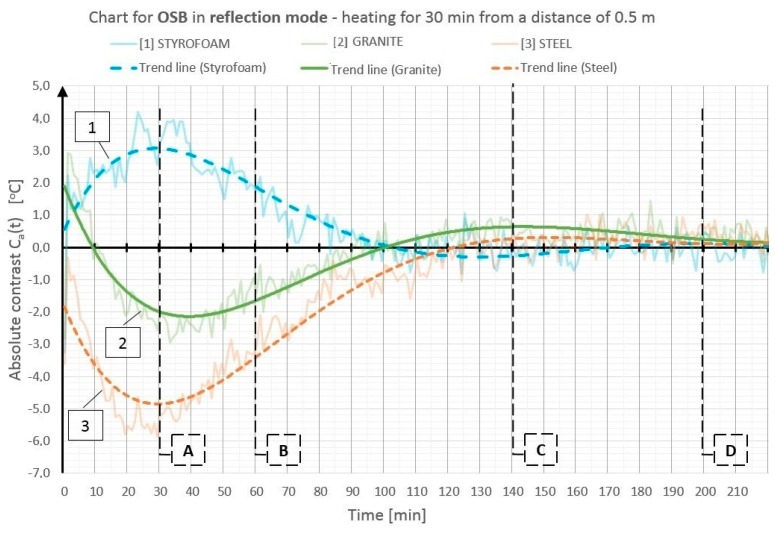
Diagram of the absolute contrast for measurement in a reflection mode for the OSB partition model with heating from a distance of 0.5 m (based on thermograms in Figure 3, row I).

**Figure 5 materials-12-00835-f005:**
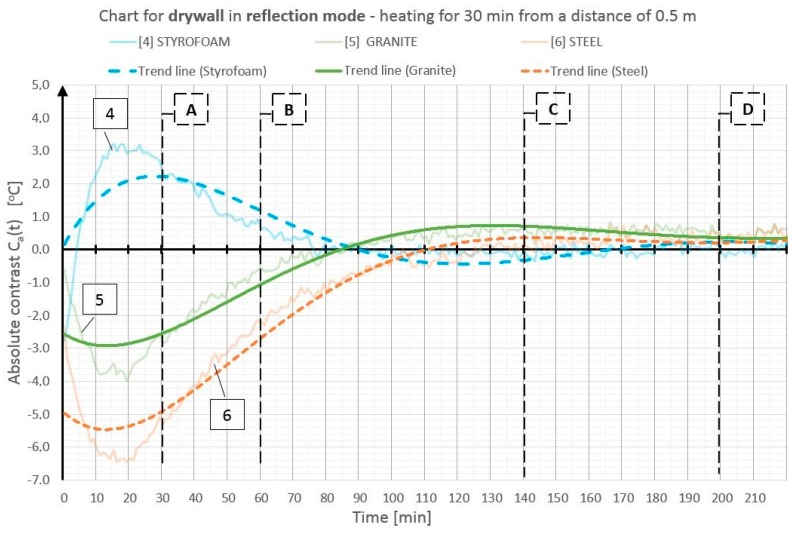
Diagram of absolute contrast for measuring the reflection mode for the GB partition model with heating from a distance of 0.5 m (based on thermograms in Figure 3, row II).

**Figure 6 materials-12-00835-f006:**
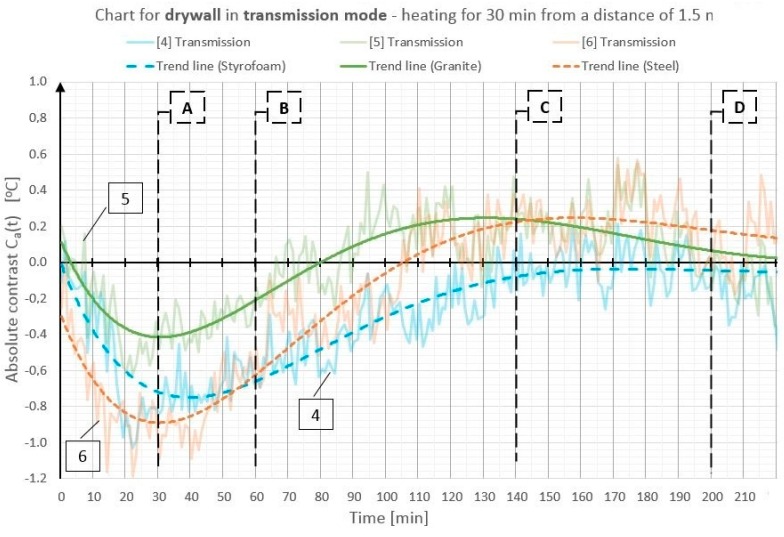
Diagram of absolute contrast for measuring in the transmission mode for the GB partition model with heating from a distance of 1.5 m (based on thermograms in Figure 3, row III).

**Figure 7 materials-12-00835-f007:**
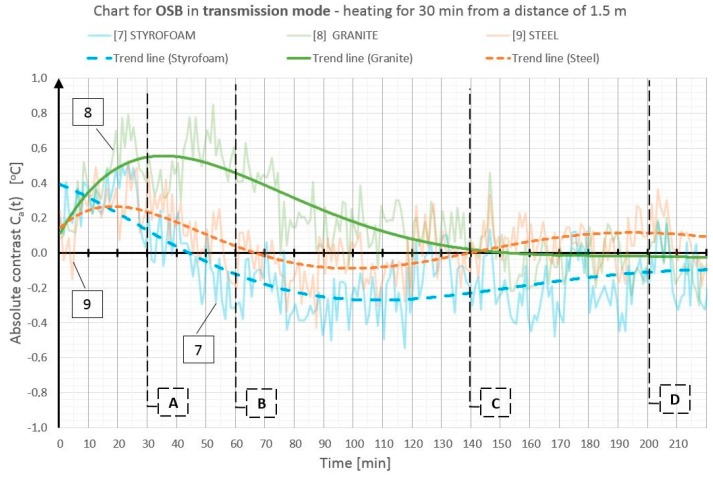
Diagram of absolute contrast for measuring in the transmission mode for the OSB partition model with heating from a distance of 1.5 m (based on thermograms in Figure 3, row IV).

**Figure 8 materials-12-00835-f008:**
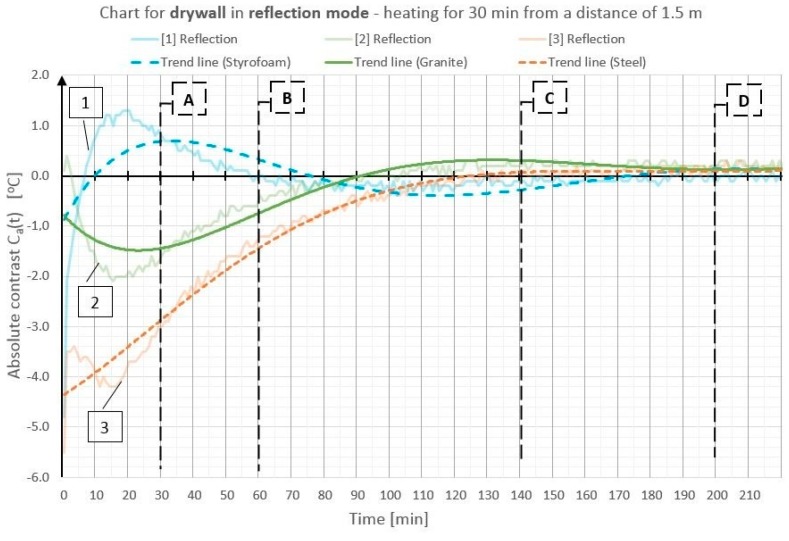
Diagram of absolute contrast for measuring in the reflection mode for a GB partition model with heating from a distance of 1.5 m (based on thermograms in Figure 3, row V).

**Figure 9 materials-12-00835-f009:**
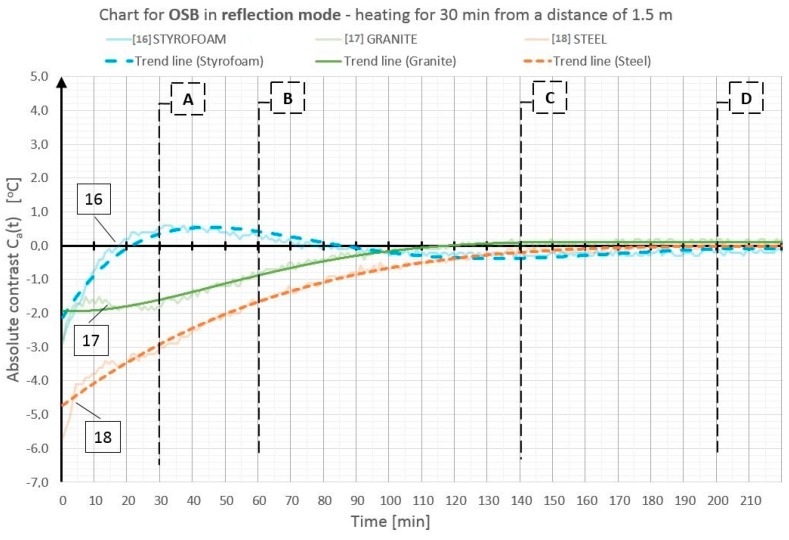
Diagram of absolute contrast for measuring in a reflection mode for the OSB partition model with heating from a distance of 1.5 m (based on thermograms in Figure 3, row VI).

**Figure 10 materials-12-00835-f010:**
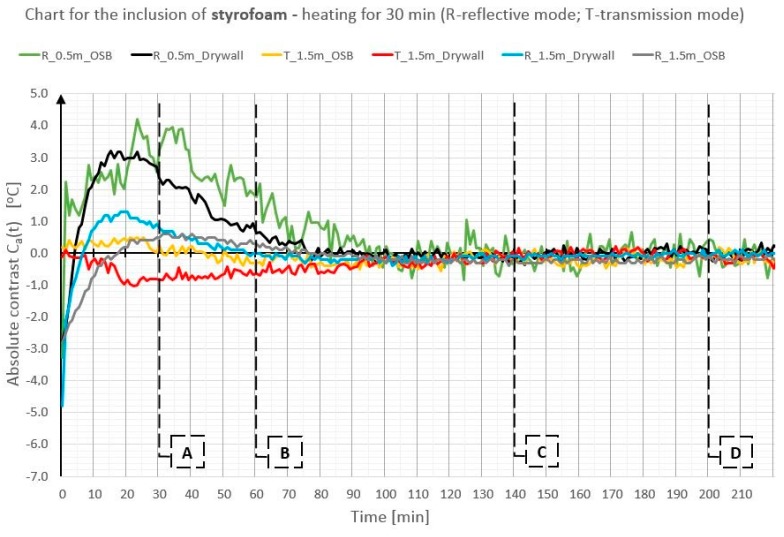
Comparison of absolute contrasts in a reflection mode and in a transmission mode for material inclusion made of Styrofoam.

**Figure 11 materials-12-00835-f011:**
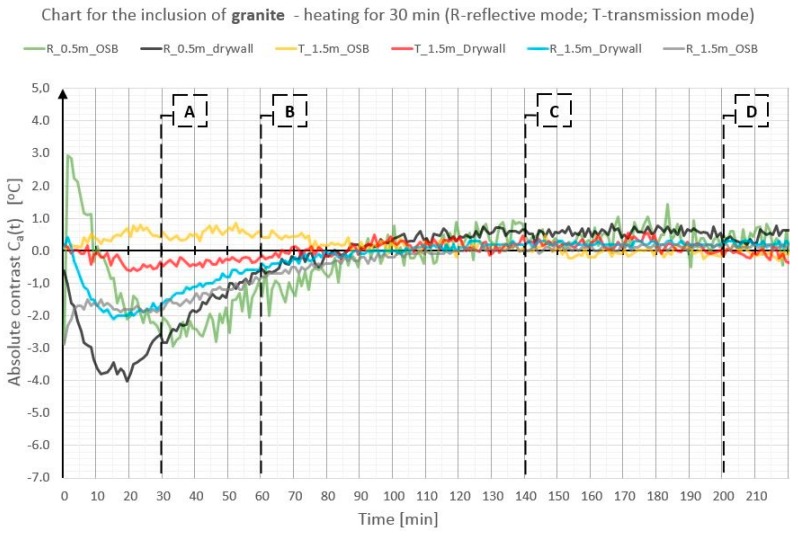
Comparison of absolute contrasts in a reflection mode and in a transmission mode for material inclusion made of granite.

**Figure 12 materials-12-00835-f012:**
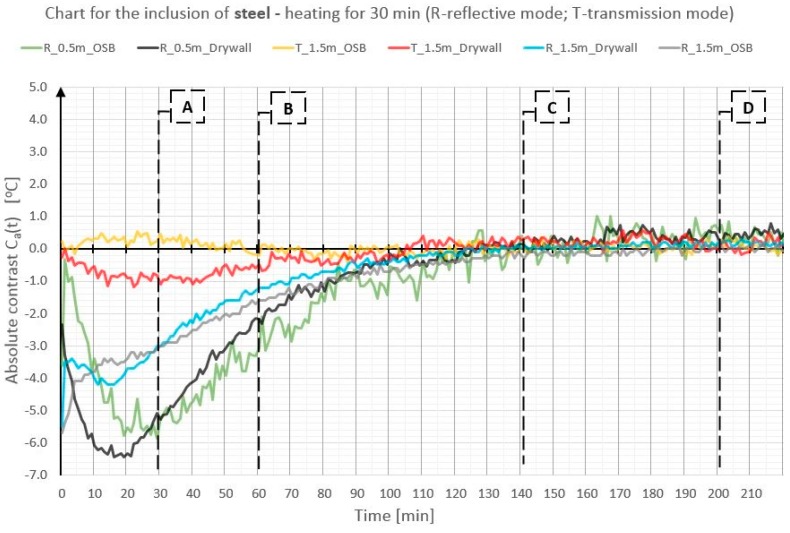
Comparison of absolute contrasts in a reflection mode and in a transmission mode for material inclusion made of steel.

**Figure 13 materials-12-00835-f013:**
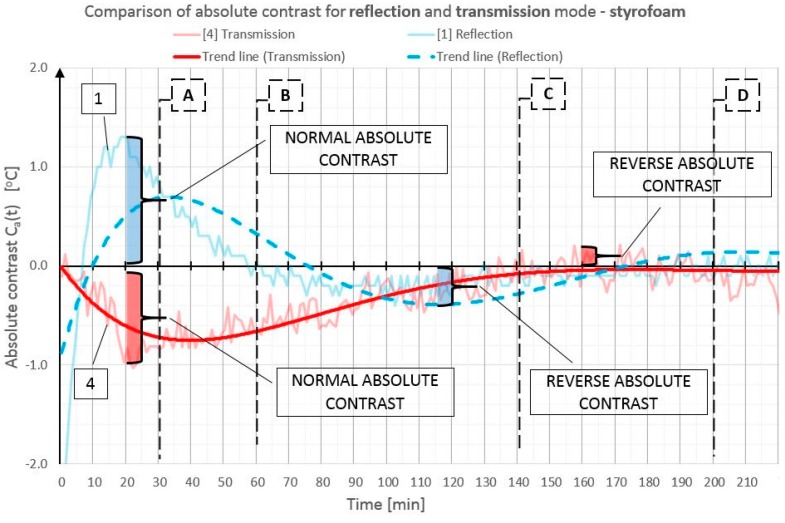
Diagram of absolute contrast in a reflection mode and in a transmission mode for Styrofoam inclusion, with marked places where normal absolute contrast and inverse absolute contrast occur (heating from a distance of 1.5 m).

**Figure 14 materials-12-00835-f014:**
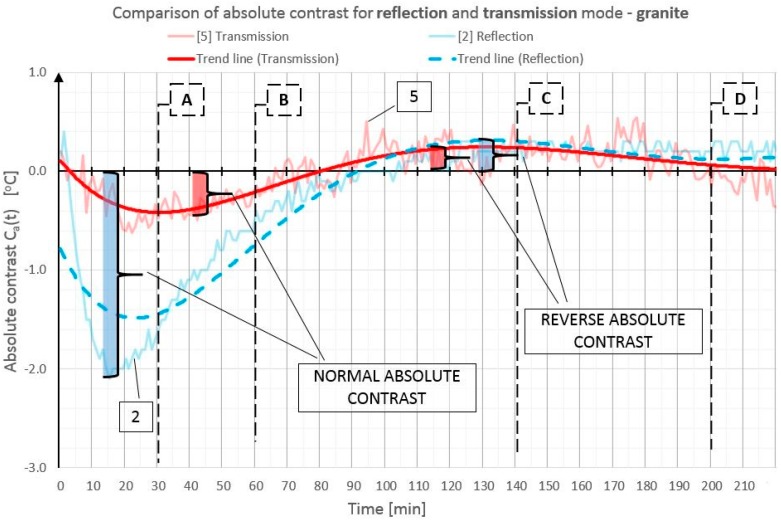
Diagram of absolute contrast in a reflection mode and in a transmission mode for granite inclusion, with marked places where normal absolute contrast and inverse absolute contrast occur (heating from a distance of 1.5 m).

**Figure 15 materials-12-00835-f015:**
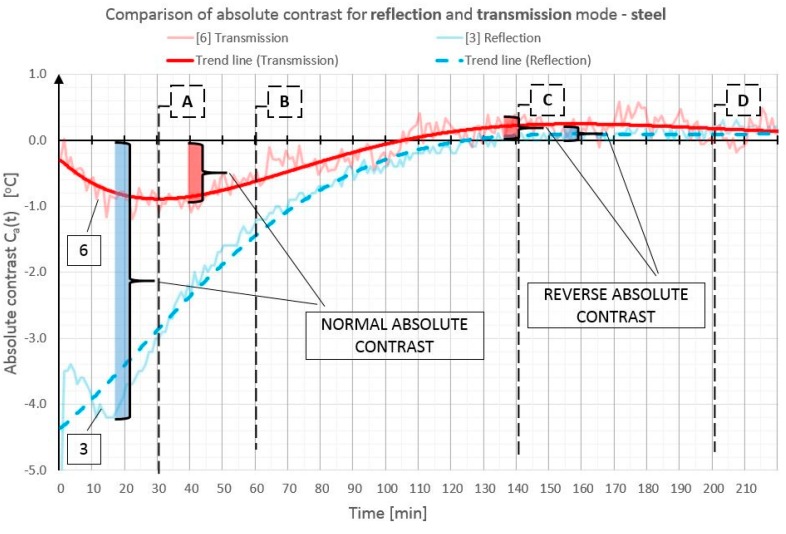
Diagram of absolute contrast in a reflection mode and in a transmission mode for steel inclusion, with marked places where normal absolute contrast and inverse absolute contrast occur (heating from a distance of 1.5 m).

**Figure 16 materials-12-00835-f016:**
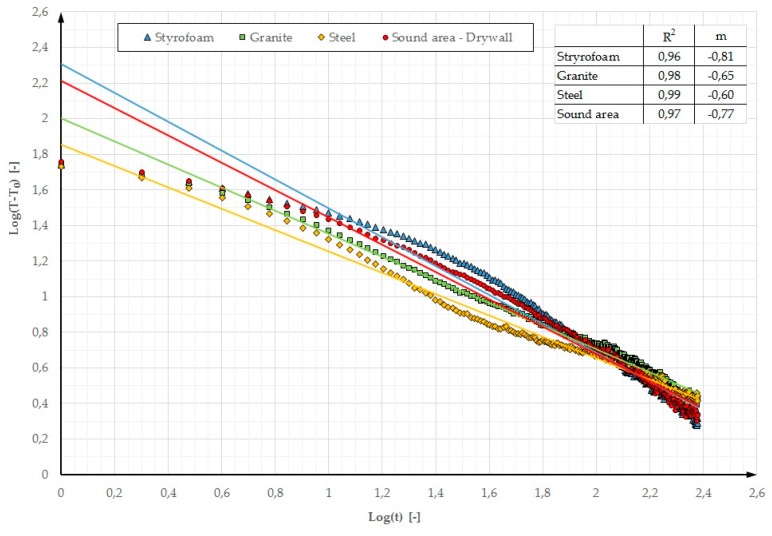
Log–log graph of the cooling phase and R^2^ and m values (stimulated period of 30 min) for the OSB partition model with heating from a distance of 0.5 m.

**Figure 17 materials-12-00835-f017:**
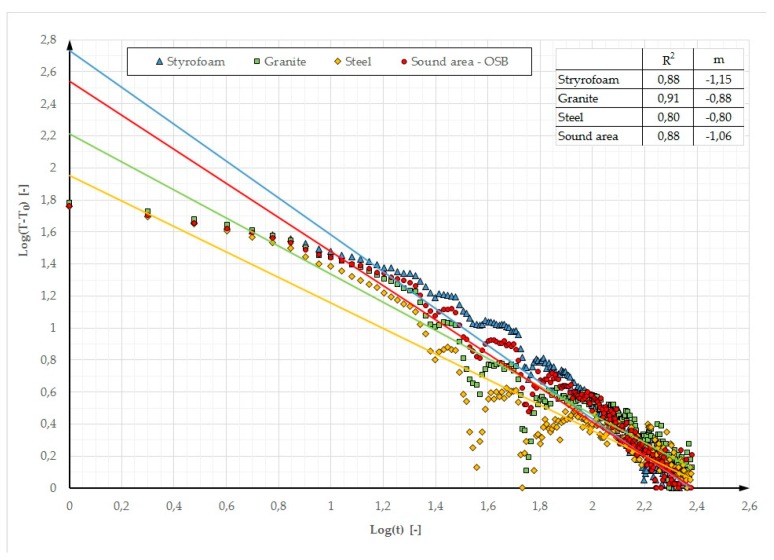
Log–log graph of the cooling phase and R^2^ and m values (stimulated period 30 min) for the GB partition model with heating from a distance of 0.5 m.

**Figure 18 materials-12-00835-f018:**
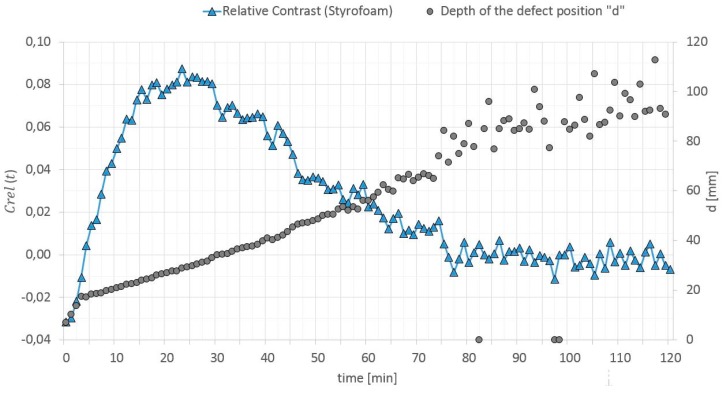
Relative contrast and defect location depth versus time (Styrofoam).

**Figure 19 materials-12-00835-f019:**
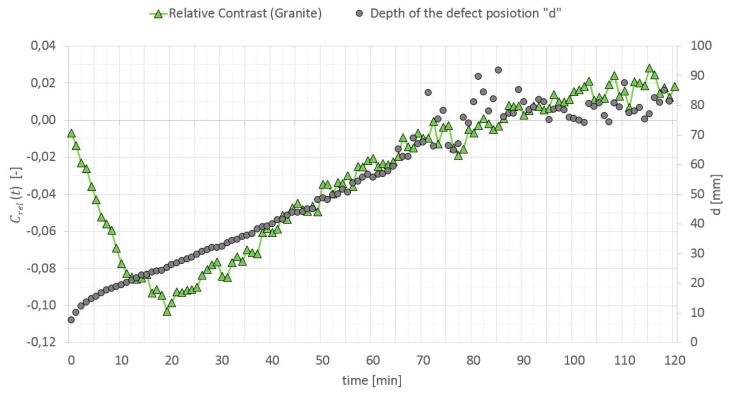
Relative contrast and defect location depth versus time (Granite).

**Figure 20 materials-12-00835-f020:**
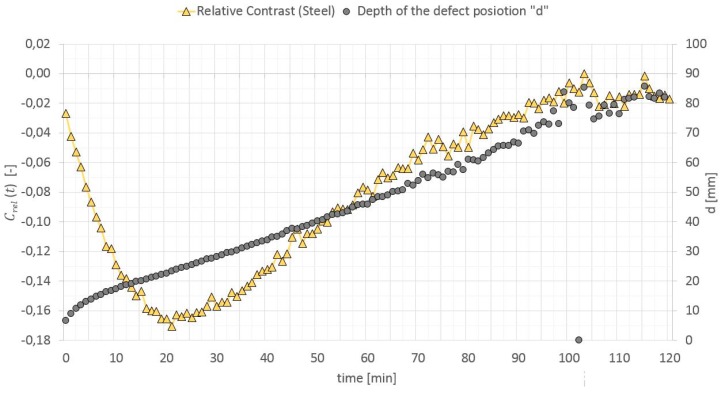
Relative contrast and defect location depth versus time (Steel).

**Figure 21 materials-12-00835-f021:**
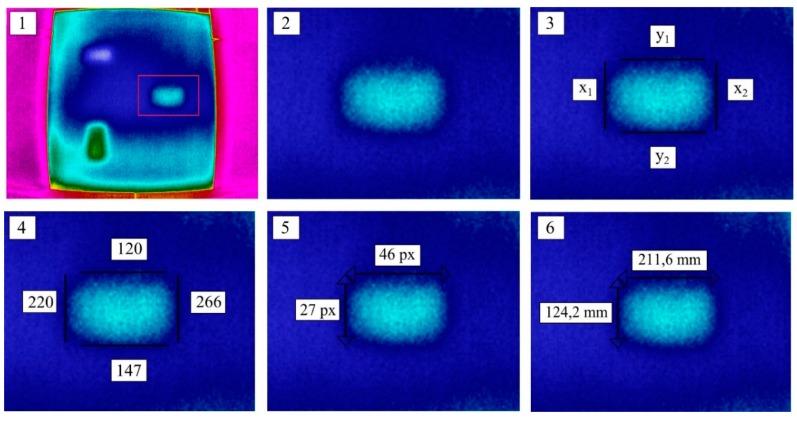
Procedure for estimating the defect size on the basis of the known size of a single pixel (4.6 mm).

**Table 1 materials-12-00835-t001:** Thermal parameters of building materials used in the experiment (values based on various literature sources).

Type of Material	Bulk Density	Specific Heat	Heat Capacity	Thermal Conductivity	Thermal Diffusivity
[–]	ρ_vol_ [kg·m^−3^]	c_w_ [J·kg^−1^·K^−1^]	C_vol_ [J·m^−3^·K^−1^]	λ [W·m^−1^·K^−1^]	a [m^2^·h^−1^]
Styrofoam	30	1460	0.04 × 10^6^	0.033	0.75 × 10^−6^
Granite	2600	920	2.39 × 10^6^	2.80	1.17 × 10^−6^
Steel	7900	500	3.95 × 10^6^	17.0	4.30 × 10^−6^
GB OSB	1000 650	1000 1700	1.00 × 10^6^ 1.11 × 10^6^	0.23 0.13	0.23 × 10^−6^ 0.12 × 10^−6^

**Table 2 materials-12-00835-t002:** Defect location depths calculated using the EDS method.

Defect	OSB, Heating from Distance 0.5 m	OSB, Heating from Distance 1.5 m	GK, Heating from Distance 0.5 m	GK, Heating from Distance 1.5 m
–	d [mm]	d [mm]	d [mm]	d [mm]
Styrofoam	21.67	27.14	22.51	22.93
Granite	24.15	20.63	22.89	21.92
Steel	19.97	22.36	21.32	19.51

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
