# Peer review of "Inverse Contrast in Non-Destructive Materials Research by Using Active Thermography"

_materials, 2019, doi:10.3390/ma12050835_

Round 1

Reviewer 1 Report

The work deals with the application of active thermography for locating materials inclusions in large-sized building. Experimental tests were carried out  to investigate different material inclusions (Styrofoam, granite and steel) and two building partition models.

General considerations:

The paper is interesting but needs major revisions for improving the results discussion.

Introduction needs to be improved:

Stepped approach has been used. Where is described this approach in introduction or in the paper ? Please refer to recent works and describe better within the paper as this technique works. Recent works have already investigated this approach:

   - Palumbo D, Galietti U, (2016), Damage Investigation in Composite Materials by Means of New Thermal Data Processing Procedures, Strain, 52(4), 276-285.

   - Palumbo D, Cavallo P, Galietti U, (2019), An investigation of the stepped thermography technique for defects evaluation in GFRP materials, NDT and E International, 102, 254-263.

In these works, the change in sign of thermal signal is explained. Please, underline the differences with the suggested papers. Moreover, these papers suggest other parameters (slope and R2) for defect detection. Please, discuss.

Please, improve the quality of Figure 1.

How did you fit the thermal contrast data ? What kind of model did you use ?

The quantitative analysis is very poor with respect to literature (see the second paper suggested). Can you improve it ?  ( by using for example the semi-contrast method). See:

-          Balageas DL, Roche JM, Leroy H, Comparative Assessment of Thermal NDT Data Processing Techniques for Carbon Fiber Reinforced Polymers, Materials Evaluation, 2017; 75(8):1019-13.

-          Giorleo G, Meola C, Comparison between pulsed and modulated thermography in glass-epoxy laminates. NDT&E International, 2002; 35:287-6.

Author Response

Dear Reviewer 1,

Thank you very much for all the comments. The answer is in the attached file.

Sincerely

Authors

Reviewer 2 Report

The main problem of the paper is that it contains neither new nor non-obvious results.  For example, the following phrases in Discussion contain no useful information: … The inclusions incorporated into the OSB and GB partition models, invisible from the outside, can be located using the active thermography method. The different thermal properties of the inclusions in comparison with those of the basic material in which these inclusions occur cause  disturbances in the temperature field distribution visible in the thermograms. The temperature is uniform in the places where there are no inclusions… But these are basics of active thermal NDT. Most of statements are well known (formula (1). In the same way, the section related to estimating defect size is obvious. This topic has been investigated much deeper in the earlier publications.

The authors state that in the course of the experiment a new kind of thermal contrast, i.e. inverse contrast, which earlier had not been described in the literature, was noticed. However, the fact that, in a one-sided test procedure, the areas over defects with lower than a host material thermal conductivity are first  ‘classically’ warmer that the surrounding non-defect areas but at longer times they may become colder. See, for example, the corresponding comment on p. 166 in an earlier work: Vavilov V.  Infrared Techniques for Materials Analysis and Nondestructive Testing. In: Infrared Methodology and Technology, ed. X.P.V. Maldague, Nondestructive Testing Monographs, vol. 7, pp.131-210. Although, the authors are right that the above-mentioned phenomenon has not been used in practice. Nevertheless, It is unclear how inverse absolute contrast can improve detection of inclusions.

In general, the paper looks more like a conference paper but not like the one suitable for a high-rank journal.

Some minor remarks are as follows.  1) Abbreviations must be explained when first mentioned. 2) The paper English is inferior in many places, for example: a) “tests to test”, b) “which consists in thermally exciting the tested element”, etc. 3) It is recommended to change the title, for instance,  for “ Inverse contrast in non-destructive materials research by using active thermography”.

With above-mentioned in mind, I do not recommend the paper for publishing.

Author Response

Dear Reviewer 2,

Thank you very much for all the comments. The answer is in the attached file.

Sincerely

Authors

Round 2

Reviewer 2 Report

The pap[er has been definitely improved.Most of authors' comments to the reviewer's remarks are appropriate. The paper can be published, perhaps, after some English editing (by the journal).